DATA RELEASE

# Long-read HiFi sequencing correctly assembles repetitive *heavy fibroin* silk genes in new moth and caddisfly genomes

Akito Y. Kawahara[1,*,†], Caroline G. Storer[1,2,†], Amanda Markee[1,3],
Jacqueline Heckenhauer[4,5], Ashlyn Powell[6], David Plotkin[1], Scott Hotaling[7],
Timothy P. Cleland[8], Rebecca B. Dikow[9], Torsten Dikow[10],
Ryoichi B. Kuranishi[11,12], Rebeccah Messcher[1], Steffen U. Pauls[4,5,13],
Russell J. Stewart[14], Koji Tojo[15] and Paul B. Frandsen[6,9,*]

1 McGuire Center for Lepidoptera and Biodiversity, Florida Museum of Natural History, University of Florida, Gainesville, FL 32611, USA
2 Pacific Biosciences, 1305 O'Brien Dr., Menlo Park, CA 94025, USA
3 School of Natural Resources and the Environment, University of Florida, Gainesville, FL 32611, USA
4 LOEWE Centre for Translational Biodiversity Genomics (LOEWE-TBG), Frankfurt 60325, Germany
5 Department of Terrestrial Zoology, Senckenberg Research Institute and Natural History Museum Frankfurt, Frankfurt 60325, Germany
6 Department of Plant and Wildlife Sciences, Brigham Young University, Provo, UT 84602, USA
7 School of Biological Sciences, Washington State University, Pullman, WA, USA
8 Museum Conservation Institute, Smithsonian Institution, Suitland, MD 20746, USA
9 Data Science Lab, Office of the Chief Information Officer, Smithsonian Institution, Washington, DC 20002, USA
10 Department of Entomology, National Museum of Natural History, Smithsonian Institution, Washington, DC, USA
11 Graduate School of Science, Chiba University, Chiba 263-8522, Japan
12 Kanagawa Institute of Technology, Kanagawa 243-0292, Japan
13 Institute for Insect Biotechnology, Justus-Liebig-University, Gießen 35390, Germany
14 Department of Biomedical Engineering, University of Utah, Salt Lake City, UT 84112, USA
15 Department of Biology, Shinshu University, Matsumoto, Nagano 390-8621, Japan

**Submitted:** 07 April 2022

\* Corresponding authors. E-mail:
kawahara@flmnh.ufl.edu;
paul_frandsen@byu.edu

† Contributed equally.

Preprint submitted at https://doi.org/10.1101/2022.06.01.494423

## ABSTRACT

Insect silk is a versatile biomaterial. Lepidoptera and Trichoptera display some of the most diverse uses of silk, with varying strength, adhesive qualities, and elastic properties. Silk fibroin genes are long (>20 Kbp), with many repetitive motifs that make them challenging to sequence. Most research thus far has focused on conserved N- and C-terminal regions of fibroin genes because a full comparison of repetitive regions across taxa has not been possible. Using the PacBio Sequel II system and SMRT sequencing, we generated high fidelity (HiFi) long-read genomic and transcriptomic sequences for the Indianmeal moth (*Plodia interpunctella*) and genomic sequences for the caddisfly *Eubasilissa regina*. Both genomes were highly contiguous (N50 = 9.7 Mbp/32.4 Mbp, L50 = 13/11) and complete (BUSCO complete = 99.3%/95.2%), with complete and contiguous recovery of silk *heavy fibroin* gene sequences. We show that HiFi long-read sequencing is helpful for understanding genes with long, repetitive regions.

**Subjects** Genetics and Genomics, Animal Genetics, Evolutionary Biology

## DATA DESCRIPTION

### Background

Many phenotypic traits across the tree of life are controlled by repeat-rich genes [1]. There are many examples, such as antifreeze proteins in fish [2], keratin in mammals, and resilin in insects [1]. Silk is a fundamental biomaterial produced by many arthropods. Silk genes are often long (>20 kilobase pairs [Kbp]) and contain repetitive motifs [3]. Accurately sequencing through repeat-rich genomic regions is critical to understanding how functional genes dictate phenotypes. However, research thus far has been unable to quantify these regions. For silk genes, this is essential because these regions control the strength and elasticity properties of silk fibers [4–6].

Lepidoptera (moths and butterflies) and their sister lineage Trichoptera (caddisflies) display some of the most diverse uses of silk, from spinning cocoons to prey capture nets and protective armorment [7]. A complete *heavy-chain fibroin* (*H-fibroin*) sequence for the model silkworm moth, *Bombyx mori*, was assembled over two decades ago using bacterial artificial chromosome libraries [8]. Recently, a combination of Oxford Nanopore Technologies (hereafter referred to as 'Nanopore') and Illumina sequencing technologies helped to generate a full *H-fibroin* sequence of *B. mori*, but large regions of the genome remain unassembled [3]. We have had similar problems with Nanopore and Illumina hybrid assemblies in caddisfly genomes e.g., [9], where we were unable to assemble complete *H-fibroin* genes despite intensive efforts for ~20 species. In these assemblies, the biggest hindrances were sequencing single strands across large repeat regions, and limited efficacy of Illumina polishing of repetitive regions in the Nanopore assembled data. Therefore, most research thus far has been limited, and has focused only on conserved N- and C-terminal regions e.g., [10]. Complete high-fidelity (HiFi) fully phased *H-fibroin* sequences are critical for advancing biomaterials discovery for insect silks.

### Context

We generated HiFi long-read genomic sequences for the Indianmeal moth (*Plodia interpunctella*, NCBI:txid58824), and the caddisfly species *Eubasilissa regina* (NCBI:txid1435191), with the Pacific Biosciences (PacBio) Sequel II system. Our goal was to recover the area of the genome that has been nearly impossible to sequence because of its repeated regions. We chose these two taxa because they represent two species with very different life histories: *Plodia interpunctella* is an important model organism in Lepidoptera whose larvae feed on various grains and stored food products and secrete large amounts of thin silken webbing at their feeding sites. They also use silk to create a cocoon during pupation [11, 12]. *Eubasilissa regina*, on the other hand, is a member of the insect order Trichoptera, whose larvae secrete silk in aquatic environments to produce protective silk cases made of broader leaf pieces from deciduous trees, cut to size [13]. These new resources not only expand our knowledge of a primary silk gene in Lepidoptera and Trichoptera, but also contribute new, high-quality genomic resources for aquatic insects and arthropods, which have thus far been underrepresented in genome biology [14–16].

## METHODS

### Sample information and sequencing

A single adult specimen of each species was sampled for inclusion in the present study. For *P. interpunctella*, we used a specimen from the PiW3 colony line at the US Department of



Agriculture laboratory in Gainesville, FL, USA. Its entire body was used for extraction, given its small size. For *E. regina*, a wild-caught female adult specimen (USNMENT01414923) from Enzan, Yamanashi, Japan (N35° 43′ 24″ E138° 50′ 33″, elevation ~4,840 ft), was used, which has been deposited in the Smithsonian National Museum of Natural History (USNM) biorepository (#AK0WP01). The head and thorax were macerated and DNA was extracted. The remainder of the tissue will be stored at the USNM biorepository.

Both specimens were flash-frozen in liquid nitrogen, and DNA was extracted using the Quick-DNA HMW MagBead Kit (Zymo Research). Extractions with at least 1 µg of high-molecular-weight DNA (>40 Kbp) were sheared, and the BluePippin system (Sage Science, Beverly, MA, USA) was used to collect fractions containing 15-Kbp fragments for library preparation.

Sequencing libraries were prepared for each species using the SMRTbell Express Template Prep Kit 2.0 (PacBio, Menlo Park, CA, USA) and following the ultra-low protocol. All sequencing was performed using the PacBio Sequel II system. For *P. interpunctella*, the genomic library was sequenced on a single 8M SMRTcell and *E. regina* was sequenced on three 8M SMRTcells, all with 30-hour movie times. For the *P. interpunctella* Iso-seq transcriptome, RNA was extracted using TRIzol (Invitrogen) from freshly dissected silk glands of caterpillars and following the manufacturer's protocol. This species has a relatively small body size than other Lepidoptera, so we waited until caterpillars reached their maximum size (during the fifth instar) before dissection, to maximize yield.

Sequencing libraries were prepared following the PacBio IsoSeq Express 2.0 Workflow and using the NEBNext Single Cell/Low Input cDNA Synthesis and Amplification Module for the SMRTbell Express Template Prep Kit 2.0. The resulting library was multiplexed and sequenced on a single Sequel II PacBio SMRT cell for 30 hours. Library preparation and sequencing was carried out at the DNA Sequencing Center at Brigham Young University (Provo, UT, USA).

Genomic HiFi reads were generated by circular consensus sequencing, where consensus sequences have three or more passes with quality values equal to or greater than 20, from the subreads.bam files and using pbccs tool (v.6.0.0) in the *pbbioconda* package (RRID:SCR_018316) [17]. Using the same *pbbioconda* package and the Iso-seq v3 tools, high quality (>Q30) transcripts were generated from HiFi read clustering without polishing.

## Genome size estimations and genome profiling

Estimation of genome characteristics, such as size, heterozygosity, and repetitiveness, were conducted using a *k*-mer distribution-based approach. After counting *k*-mers with K-Mer Counter (KMC) v.3.1.1 (RRID:SCR_001245) and a *k*-mer length of 21 (–m 21), we generated a histogram of *k*-mer frequencies with KMC transform histogram [18]. We then generated genome *k-mer* profiles on the *k*-mer count histogram using the GenomeScope 2.0 web tool (RRID:SCR_017014) [19], with the *k*-mer length set to 21 and the ploidy set to 2.

## Sequence assembly and analysis

For both genomes, reads were then assembled into contigs using the assembler Hifiasm v0.13-r307 (RRID:SCR_021069) with aggressive duplicate purging enabled (option –l 2) [20]. The primary contig assembly was used for all downstream analyses. Genome contiguity was measured using assembly_stats.py [21] and genome completeness was determined using BUSCO v.5.2.2 (RRID:SCR_015008) [22] and the obd10 reference Endopterygota.



**Table 1.** Specimen accession and data type information.

| Species | BioProject | BioSample | Assembly | SRA | Sequence type |
|---|---|---|---|---|---|
| *Plodia interpunctella* | PRJNA741212 | SAMN20990134 | NA | SRR15699974 | Transcriptome |
| *Plodia interpunctella* | PRJNA741212 | SAMN19857939 | JAJAFS000000000 | SRR15658214 | Genome |
| *Eubasilissa regina* | PRJNA741212 | SAMN20522324 | JAINEB000000000 | SRR15651978 | Genome |

NA: Not applicable; SRA: Sequence Read Archive.

Contamination in the genome was assessed by creating taxon-annotated GC-coverage plots using BlobTools v1.0 (RRID:SCR_017618) [23]. First, assemblies were indexed using samtools faidx, then HiFi reads were mapped back to the indexed assemblies using minimap2 (RRID:SCR_018550) [24] with –ax asm20. The resulting bam files were sorted with samtools sort. Taxonomic assignment was performed via Megablast and using the NCBI nucleotide database [25] with parameters –outfmt 6 qseqid staxids bitscore std' –max_target_seqs 1 –max_hsps 1-e value 1e-25. BlobPlots were created by making a blobtools database from the assembly file, BLAST results, and mapping results using blobtools create and plots were created using blobtools plot.

## Genome statistics

All samples, raw sequence reads, and assemblies were deposited to GenBank [26] (Table 1). We generated 35.7 gigabase pairs (Gbp; 41× coverage) and 15.7 Gbp (44× coverage) of PacBio HiFi sequence for *E. regina* and *P. interpunctella*, respectively. We assembled those reads into two contiguous genome assemblies. The assembly for *E. regina* has the highest contig N50 of any Trichoptera genome assembly to date. It contains 123 contigs, a contig N50 of 32.4 Mbp, GC content of 32.68%, and a total length of 917,780,411 basepairs (bp). GenomeScope 2.0 estimated a genome size of 854,331,742 bp with 75.3% unique sequence [27]. Despite recent analyses showing no evidence of whole-genome duplication in caddisflies [9], the findings in this study may be an indication of tetraploidy. Future research should be done to further examine these patterns.

The *P. interpunctella* assembly represents a substantial improvement to existing, publicly available genome assemblies (Tables 2 and 3). After contaminated contigs were removed (three contigs contaminated with *Wolbachia* were identified), the resulting assembly comprises 118 contigs with a cumulative length of 300,731,903 bp. It has a contig N50 of 9.7 Mbp and a GC content of 35.41%. The genome size estimated by GenomeScope 2.0 was 275,458,564 bp, with 87.1% unique sequence [28].

## Heavy-chain fibroin gene annotation

We extracted *H-fibroin* silk genes from both the *P. interpunctella* and *E. regina* assemblies. For *P. interpunctella*, we also searched existing, short-read based assemblies. We downloaded two short-read based genome assemblies for *P. interpunctella*, GCA_001368715.1 and GCA_900182495.1 from NCBI [29]. Since the internal region of *H-fibroin* is repetitive, the more conserved N- and C-termini amino acids were blasted against the genomes with tblastn (RRID:SCR_011822) [29]. For *P. interpunctella*, we used the terminal sequences published in [30] and for *E. regina*, we used the terminal sequences published in [5]. We then extracted the sequences and 500 bp of flanking regions from the assembly and annotated them using Augustus v.3.3.2 [31]. Spurious introns (those that did



**Table 2.** Assembly genome statistics for the species sampled in this study.

| Parameter | *Plodia. interpunctella* | *Eubasilissa regina* | *Plodia interpunctella* | *Plodia interpunctella* |
|---|---|---|---|---|
| Reference | This study | This study | GCA_001368715.1 | GCA_900182495.1 |
| Platform | PacBio Sequel II | PacBio Sequel II | Illumina MiSeq/HiSeq | Illumina MiSeq/HiSeq |
| Coverage (×) | 44 | 41 | 100 | 50 |
| Total ungapped length (bp) | 300,731,903 | 917,780,411 | 364,621,386 | 364,623,808 |
| Total gapped length (bp) | NA | NA | 382,235,502 | 381,952,380 |
| Number of scaffolds | NA | NA | 7743 | 10,542 |
| Scaffold N50 | NA | NA | 5,094,612 | 1,270,674 |
| Scaffold L50 | NA | NA | 23 | 75 |
| Number of contigs | 118 | 123 | 17,717 | 17,725 |
| Contig N50 | 9,707,027 | 32,427,664 | 302,097 | 298,497 |
| Contig L50 | 13 | 11 | 314 | 319 |
| GC content (%) | 35.41 | 32.68 | 35.1 | 35.1 |
| Shortest Contig (bp) | 452 | 15,452 | 258 | 258 |
| Longest Contig (bp) | 13,555,736 | 57,864,696 | 2,314,344 | 2,314,344 |
| Median Contig (bp) | 161,724 | 36,760 | 1,714 | 1,719 |
| Mean Contig (bp) | 2,548,575 | 7,401,455 | 20,580 | 20,571 |

**Table 3.** Genome completeness by sample studied. Values shown are BUSCO scores for the Endopterygota ODB10 data set.

| Parameter | *Plodia interpunctella* | *Eubasilissa regina* | *Plodia interpunctella* | *Plodia interpunctella* |
|---|---|---|---|---|
| Reference | This study | This study | GCA_001368715 | GCA_900182495 |
| Complete BUSCOs | 2110 | 2021 | 2103 | 2105 |
| Complete and single copy | 2097 | 2013 | 2074 | 2077 |
| Complete and duplicated | 13 | 14 | 29 | 28 |
| Fragmented | 5 | 63 | 10 | 8 |
| Missing | 9 | 34 | 11 | 11 |
| Total groups searched | 2124 | 2124 | 2124 | 2124 |
| % complete | 99.3 | 95.2 | 99.0 | 99.1 |

not affect reading frames and were not supported by transcript evidence) were manually removed. Annotated sequences are provided in the *Gigascience* GigaDB repository [32].

We recovered full-length *H-fibroin* sequences in both genomes. To our knowledge, the only other previously published full-length lepidopteran *H-fibroin* sequence was from a BAC library-based sequence of the model organism, *B. mori*. We compared our assembly of the *P. interpunctella H-fibroin* sequence with that from a previously published Illumina-based genome assembly of the same species (Figure 1). Where the Illumina-based assembly only recovered the conserved terminal regions and a small number of repetitive elements, our assembly recovered the full-length gene, including the full complement of repetitive motifs (Figures 1, 2). Specifically, the *P. interpunctella* genome had a *H-fibroin* sequence that was 14,866 bp (whole gene with introns; 4,714 amino acids), and a molecular weight of 413,334.41 Da. For *E. regina*, we recovered the full-length sequence of *H-fibroin*, which was 25,250 bp (whole gene with introns; 8,386 amino acids), and a molecular weight of 815,864.95 Da, with repeated regions (Figure 3).

The recovery of this *H-fibroin* sequence marks the third complete, published *H-fibroin* sequence in Trichoptera [33, 34]. Our work shows that high quality, long-read sequencing can be used to successfully assemble difficult regions of non-model organisms without the use of expensive and tedious BAC methods. While our study is focused on the repetitive silk gene, *H-fibroin*, these results likely extend to other long, repetitive proteins that have previously proven difficult to assemble.

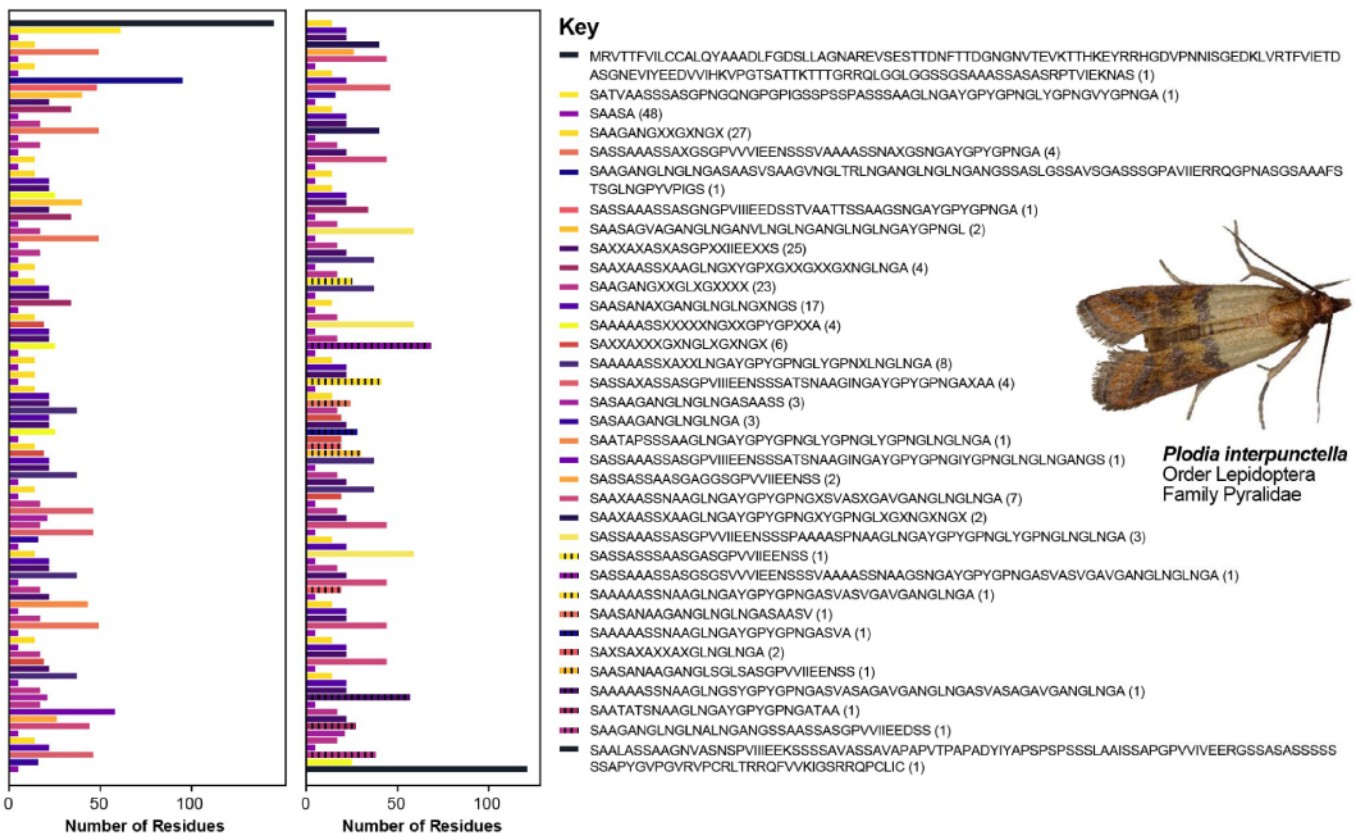

**Figure 1.** Length of assembled *heavy fibroin* (*HFib*) gene for *Plodia interpunctella* using two approaches. Top shows HiFi assembly, bottom shows Illumina assembly. In the HiFi genome, we recovered the entire length of the sequence, but in the Illumina assembly we could not assemble the genome through the repetitive region.

**Figure 2.** Schematic of the identity and ordering of repeat motifs in *Plodia interpunctella*. On the right panel are the repetitive units with the *N*-terminus at the beginning and the *C*-terminus at the end. The numbers in parentheses refer to the number of times that particular motif is repeated across the gene. The color corresponds with the ordering of the repeats shown on the left. The gene is split into two panels, starting in the left panel and continuing in the right panel. "X" indicates a variable site.

## Genome annotation

For structural annotations of the genomes, we masked and annotated repetitive elements using RepeatMasker (RRID:SCR_012954) [35] after identifying and classifying them *de novo*



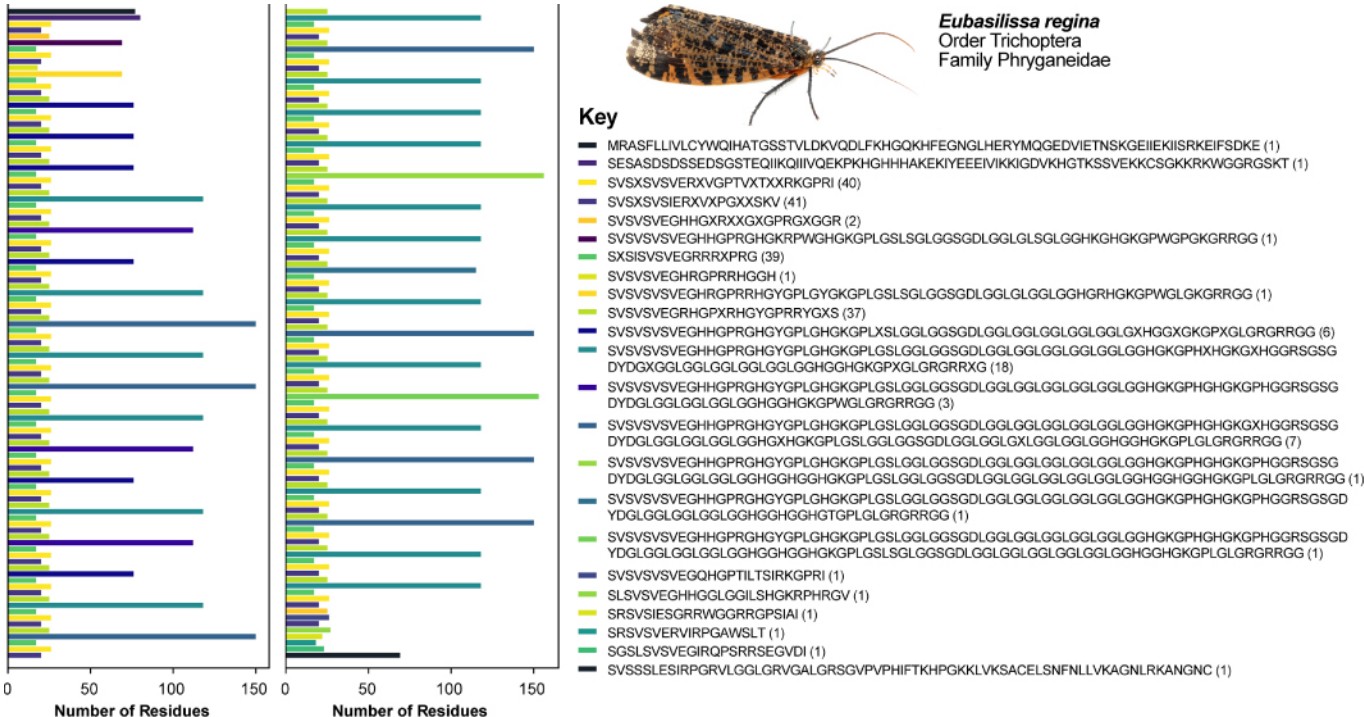

**Figure 3.** Schematic of the identity and ordering of repeat motifs in *Eubasilissa regina*. On the right panel are the repetitive units with the *N*-terminus at the beginning and the *C*-terminus at the end. The numbers in parentheses refer to the number of times that particular motif is repeated across the gene. The color corresponds with the ordering of the repeats shown on the left. The gene is split into two panels, starting in the left panel and continuing in the right panel.

with RepeatModeler2 (RRID:SCR_015027) [36] following a previously published protocol [37]. For species-specific gene model training, we used BUSCO v.4.1.4 [22] with the Endopterygota odb10 core ortholog sets [38], with the –long option in genome mode. In addition, we predicted genes with the homology-based gene prediction GeMoMaPipeline of GeMoMa v1.6.4 (RRID:SCR_017646) [39, 40] using previously published genomes. For *E. regina* we used the genome of *Agypnia vestita* (JADDOH000000000.1) [41] and for *P. interpunctella* we used the genome of *Bombyx mori* (GCF_014905235) as a reference. We then used the MAKER v3.01.03 pipeline (RRID:SCR_005309) [42] to generate additional *ab initio* gene predictions with the proteins predicted from GeMoMa for protein homology evidence and the Augustus-generated gene prediction models from BUSCO for gene prediction. For expressed sequence tag evidence, we used the transcriptome of *Ptilostomis semifasciata* (111015_I297_FCD05HRACXX_ L1_INSbttTHRAAPEI-17 [43]) for *E. regina* and Iso-seq data for *P. interpunctella*. Evidence used in Maker and the Maker config files can be found in the *Gigascience* GigaDB repository [32].

To add functional annotations to the predicted proteins, we blasted the predicted proteins against the ncbi-blast protein database using BlastP (RRID:SCR_001010) in blast.2.9 [29] with an *e*-value cutoff of $10^{-4}$ and –max_target_seqs set to 10 [32]. We then used the command line version of Blast2GO v.1.4.4 (RRID:SCR_005828) [44] to assign functional annotation and GO terms.



## DATA VALIDATION AND QUALITY CONTROL

In addition to full-length *H-fibroin* sequences, we recovered a high number of single-copy orthologs in each genome with BUSCO. The *E. regina* genome contained 95.2% of an Endopterygota core gene collection (comprising 2124 genes), indicating an almost complete coverage of known single-copy orthologs in the coding fraction. While the number of single-copy orthologs recovered in the new *P. interpunctella* genome was similar to earlier published genomes (99.3% of the Endopterygota core gene collection, 99.1% of the Lepidoptera core gene collection), the full-length sequence of *H-fibroin* only recovered in the HiFi based genome gives some indication of how other portions of the genome may have assembled. Following contamination screening by NCBI, we filtered out three instances of *Wolbachia* contamination in the *P. interpunctella* genome. BlobPlots for both genomes revealed low levels of contamination (Figures 4 and 5).

### Structural and functional annotation

A total of 56.26% of the *E. regina* genome was classified as repetitive (54.2% interspersed repeats). More than half of the interspersed repeats, 29.87%, could not be classified by comparison with known repeat databases, and therefore may be specific for Trichoptera. Of the classified repeats, retroelements were the most abundant, comprising 15.35% (of which 14.55% are long interspersed nuclear elements [LINEs]) of the genome. The relatively high proportion of repetitive sequence supports previous studies, which suggest that repetitive element expansion occurred in lineages of tube case-making caddisflies, such as the closely related genera *Agrypnia* and *Hesperophylax* [9, 41]. In contrast, a total of 31.94% of the *P. interpunctella* genome assembly was masked as repeats. A 23.04% of the annotated repeats were interspersed repeats. Details on the repeat classes are given in the *Gigascience* GigaDB repository [32].

Genome annotations resulted in the prediction of 16,937 and 60,686 proteins in *P. interpunctella* and *E. regina*, respectively. Of the annotated proteins, for *E. regina*, 28,358 showed significant sequence similarity to entries in the NCBI non-redudant database; of those, 12,550 were mapped to Gene Ontology (GO) terms, and 5652 were functionally annotated with Blast2GO. For *P. interpunctella*, 16,349 were verified by BLAST, 12,410 were mapped to GO terms, and 9,711 were functionally annotated in Blast2GO.

The major biological process found in the two genomes were cellular (*E. regina*: 2326 genes; *P. interpunctella:* 4725 genes) and metabolic (*E. regina*: 2454 genes; *P. interpunctella:* 3699 genes) processes. Binding (*E. regina*: 2382 genes; *P. interpunctella:* 4405 genes) and catalytic activity (*E. regina*: 2778 genes; *P. interpunctella:* 3893 genes) were the largest subcategories in molecular function. Regarding the cellular component category, most genes were assigned to the cell (1553 genes) and membrane (1491 genes) subcategory in *E. regina,* and to the cellular anatomical entity subcategory in *P. interpunctella* (5602 genes). The major biological process found in both genomes were cellular and metabolic processes.

### REUSE POTENTIAL

We provide a complete genome of two species of silk-producing insects in the superorder Amphiesmenoptera; the moth *P. interpunctella* and the caddisfly *E. regina*. We also recover the difficult-to-sequence repetitive regions of both genomes with HiFi sequencing. *P. interpunctella* is currently being developed in multiple laboratoriess as a model organism, and this genome assembly will facilitate molecular genetics research on this species.

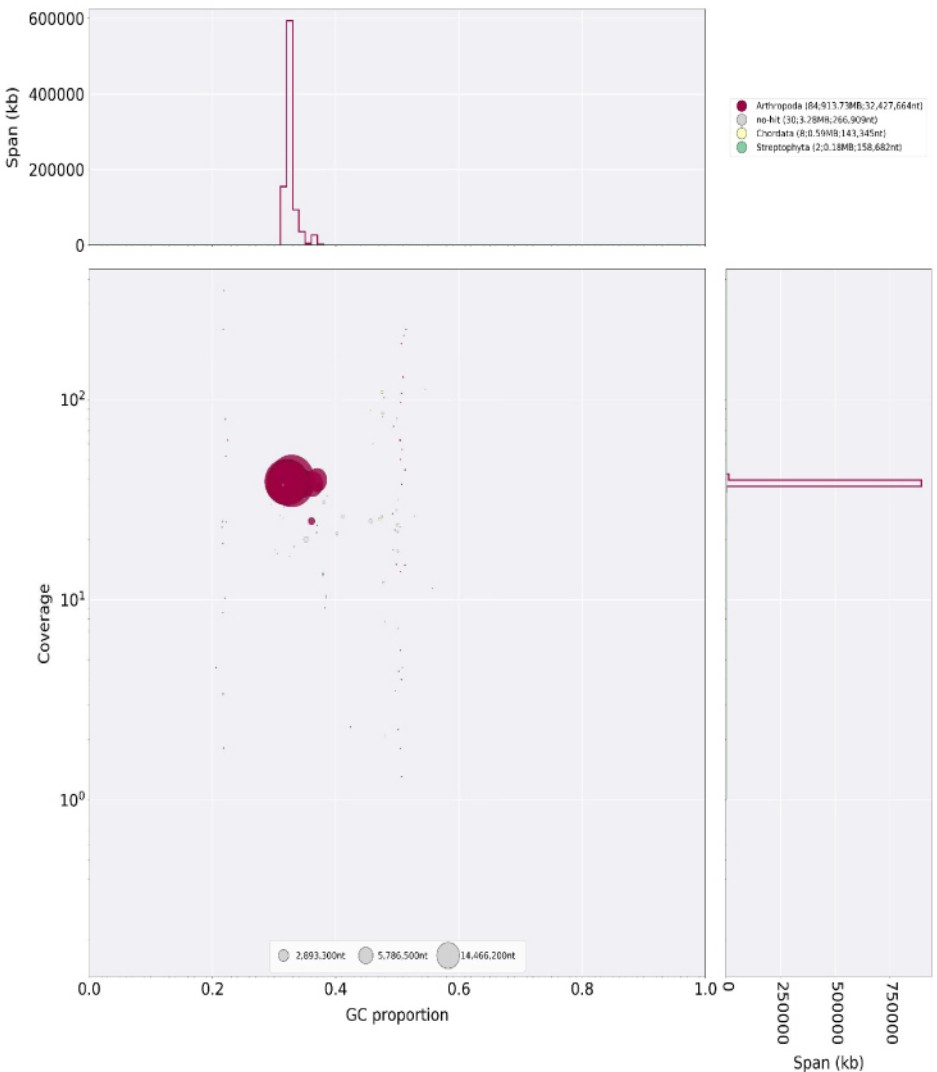

**Figure 4.** BlobPlot for *Eubasilissa regina*.

We show that PacBio HiFi sequencing allows accurate generation of repetitive protein-coding regions of the genome (silk *fibroins*), and this probably applies to other similarly repetitive regions of the genome. For Trichoptera, there are only four other HiFi genome assemblies available on Genbank, only one of which has been published [45]. Insects have largely been neglected (relative to their total species diversity) in terms of genome sequencing efforts [15, 16], which is especially true for aquatic insects [14]. These data serve as the first step to study the evolution of adhesive silk in Amphiesmenoptera, which is an innovation beneficial for survival in aquatic and terrestrial environments. Finally, the Iso-seq data that we provide serve as useful resources for the translational aspects of silk. These data provide information on how Amphiesmenoptera genetically

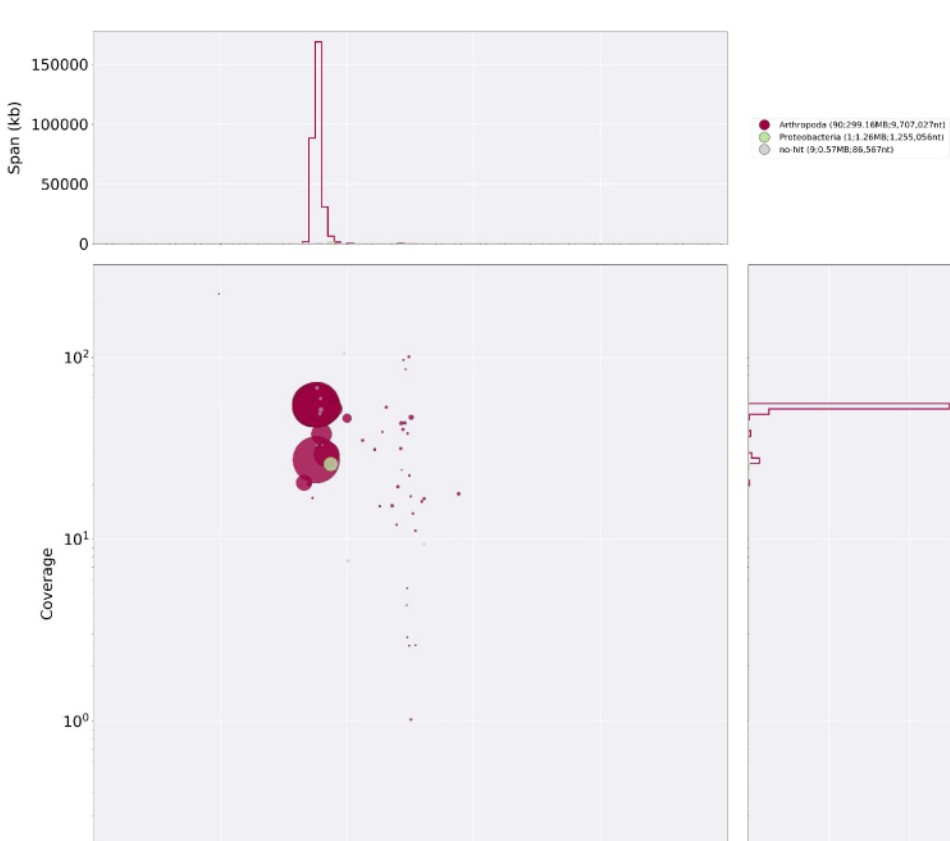

**Figure 5.** BlobPlot for *Plodia interpunctella*.

modulate and regulate different silk properties, which allows them to use silk for different purposes, such as for nets, cases, and cocoons in both terrestrial and aquatic environments.

## AVAILABILITY OF SOURCE CODE AND REQUIREMENTS

All custom-made scripts used in this study are available on GitHub.

- Project name: Silk gene visualization
- Project home page: https://github.com/AshlynPowell/silk-gene-visualization/tree/main
- Operating system(s): Platform independent
- Programming language: Python
- Other requirements: none
- License: MIT

## DATA AVAILABILITY

Raw sequence data, genome assemblies, and sample information are all available from NCBI under Bioproject number PRJNA741212. Individual accessions can be found in Table 1. Snapshots of the code and supporting data are available in GigaDB [32], including assemblies and annotations for *P. interpunctella* [46] and *E. regina* [47].

## DECLARATIONS
## LIST OF ABBREVIATIONS

bp: base pair; Gbp: gigabase pair; HiFi: high fidelity; Kbp: kilobase pair; Nanopore: Oxford Nanopore Technology; PacBio: Pacific Biosciences.

## ETHICAL APPROVAL

Not applicable.

## CONSENT FOR PUBLICATION

Not applicable.

## COMPETING INTERESTS

The authors declare that they have no competing interests.

## FUNDING

This study was funded by the Smithsonian National Museum of Natural History Global Genome Initiative (GGI-Peer-2018-182) to TPC, RD, TD, AYK; the Smithsonian Museum Conservation Institute Federal; and Trust funds to TPC and PBF. A grant from the University of Florida Research Opportunity Seed Fund internal award (number AWD06265) was awarded to principal investigators AYK and CGS. The LOEWE Centre for Translational Biodiversity Genomics (TBG) is funded by the Hessen State Ministry of Higher Education, Research and the Arts (HMWK), which financially supported JH and SUP. SH was supported by National Science Foundation award #OPP-1906015.

## AUTHORS' CONTRIBUTIONS

AYK: Designed project, collected samples, provided computational resources, manuscript writing. CGS: Designed project, data analysis, manuscript writing. AM: Sample preparation, manage colonies, manuscript writing. JH: Data analysis, manuscript writing. AP: Data analysis, manuscript writing. DP: Data file management, manuscript writing. SH: Visualization, manuscript writing. TPC: Grant writing, manuscript writing. RBD: Grant writing, manuscript writing. TD: Grant writing, manuscript writing. RBK: Collected samples, manuscript writing. RM: Helped with sample preparation, manage colonies, manuscript writing. SUP: provided computational resources, manuscript writing. RJS: Grant writing, manuscript writing. KT: Collected samples, manuscript writing. PBF: Designed project, collected samples, conducted analyses, provided computational resources, manuscript writing.

## ACKNOWLEDGEMENTS

We thank Brigham Young University, the University of Florida, and LOEWE-Centre for Translational Biodiversity Genomics (TBG) High Performance Computing clusters for providing the computational resources needed to complete this study.

## AUTHORS' INFORMATION

AM: Graduate student at the University of Florida, School of Natural Resources and Environment. Studies variation in lepidopteran silk production.

AP: Undergraduate student at Brigham Young University. Studies the genomics of caddisfly silk.

AYK: Associate Curator at the University of Florida. Works on Lepidoptera, genomics, and evolution.

CGS: Formerly Assistant Scientist at the University of Florida. Expert on silks and Lepidoptera. Currently works at Pacific Biosciences.

DP: Project manager at the Florida Museum of Natural History, University of Florida. Works on Lepidoptera systematics and evolution.

JH: Postdoctoral researcher at LOEWE TBG, interested in biodiversity genomics, comparative genomics, evolution and phylogenomics.

KT: Professor of Shinshu University, Japan, Trichoptera specialist.

PBF: Assistant Professor of Genetics, Genomics, and Biotechnology at Brigham Young University, specialist on the genomics of caddisflies and their silk.

RBD: Data Scientist at Smithsonian National Museum, Washington, DC.

RJS: Professor of Bioengineering, University of Utah, specialist on the biomechanics of caddisfly silk.

RBK: Guest Professor of Kanagawa Institute of Technology, specialist of Trichoptera.

RM: Biological Scientist at the University of Florida researching molecular biology and genetics of organisms.

SH: Postdoctoral Research Associate at Washington State University and a specialist in insect genomics.

SUP: Entomologist at Senckenberg Research Institute and Natural History Museum Frankfurt and Justus-Liebig-University, interested in the evolution and ecology of freshwater insects, especially Trichoptera.

TD: Researcher and curator at Smithsonian National Museum of Natural History, Washington, DC, Works on biodiversity of flies and is curator for aquatic insects.

TPC: Physical Scientist at the Smithsonian Museum Conservation Institute.

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
