## [Reviewer Report]

Reviewer name and names of any other individual's who aided in reviewer Peter MulhairDo you understand and agree to our policy of having open and named reviews, and having your review included with the published papers. (If no, please inform the editor that you cannot review this manuscript.)YesIs the language of sufficient quality?YesPlease add additional comments on language quality to clarify if needed
This manuscript is clear and concise. However, there are some issues with consistency in species names used throughout the manuscript. First, on line 99 Eubasilissa regina should be italicised. Secondly, I would recommend after the initial use of the full names of the species (Plodia interpunctella and Eubasilissa regina) that these be referred to as P. interpunctella and E. regina in the rest of the text. There is inconsistent use of full species names, shortened species names and genus name alone which may cause confusion. Please read through and correct these inconsistencies throughout the manuscript text.Are all data available and do they match the descriptions in the paper? YesAdditional CommentsAre the data and metadata consistent with relevant minimum information or reporting standards? See GigaDB checklists for examples <a href="http://gigadb.org/site/guide" target="_blank">http://gigadb.org/site/guide</a>NoAdditional CommentsMissing items from the metadata checklist include (1) Coding gene annotations (GFF), Coding gene nucleotide sequences and Coding gene translated sequences (fasta) and (2) Full (not summary) BUSCO results output files (text).Is the data acquisition clear, complete and methodologically sound?YesAdditional CommentsIs there a specific reason why fifth instar larvae were used for RNA sequencing of silk glands of P. interpunctella? If this stage is biologically important than it may be worth stating why this specific stage is used.Is there sufficient detail in the methods and data-processing steps to allow reproduction?YesAdditional CommentsHowever, the code used for Heavy fibroin gene annotation could be made publicly available to enable reproducibility of this analysis (using other species for example or to annotate other repeat rich genes). This could be uploaded to the rest of the relevant code at https://github.com/AshlynPowell/silk-gene-visualizationIs there sufficient data validation and statistical analyses of data quality? YesAdditional CommentsIs the validation suitable for this type of data?YesAdditional CommentsIs there sufficient information for others to reuse this dataset or integrate it with other data?YesAdditional CommentsOne point worth making is that on Line 161 you state that "The assembly for E. regina is the most contiguous Trichoptera genome assembly to date.". However, there are currently 3 chromosome level assemblies available for Trichoptera on NCBI. I would recommend removing this statement, or changing it by also pointing to these other genomes available.Any Additional Overall Comments to the AuthorThis work was carried out to a very high quality and I am particularly happy to see more high quality genomic and transcriptomic data for these groups of insects. I also think that annotation of the Heavy fibroin genes is of particular importance and relevance to researchers interested in silk evolution and evolution and annotation of repeat rich proteins.RecommendationMinor Revision

---

## [Reviewer Report]

Reviewer name and names of any other individual's who aided in reviewer Reuben W NowellDo you understand and agree to our policy of having open and named reviews, and having your review included with the published papers. (If no, please inform the editor that you cannot review this manuscript.)YesIs the language of sufficient quality?YesPlease add additional comments on language quality to clarify if needed
Are all data available and do they match the descriptions in the paper? NoAdditional CommentsI wasn't able to access the data with the FTP link provided.Are the data and metadata consistent with relevant minimum information or reporting standards? See GigaDB checklists for examples <a href="http://gigadb.org/site/guide" target="_blank">http://gigadb.org/site/guide</a>YesAdditional CommentsIs the data acquisition clear, complete and methodologically sound?YesAdditional CommentsIs there sufficient detail in the methods and data-processing steps to allow reproduction?YesAdditional CommentsIs there sufficient data validation and statistical analyses of data quality? YesAdditional CommentsIs the validation suitable for this type of data?YesAdditional CommentsIs there sufficient information for others to reuse this dataset or integrate it with other data?YesAdditional CommentsAny Additional Overall Comments to the AuthorA very nice piece of work, I have only a few minor comments:

- Line 140: "with the k-mer length set to 1" - do you mean 21?
- Line 164: great that you provide a link to the GenomeScope html but I recommend to add these kmer plots as additional supplemental figures, they are extremely useful. Just a screenshot of the GenomeScope plot would be fine. 
- Line 164: in relation to the kmer distributions, in fact both plots look a little bit multimodal to me... especially the Eubasilissa, with peaks at 1n (20x), 2n (40x) and 4n (~80x) coverage. This might indicate tetraploidy, which might explain the large increase in genome span and gene number for this species too. You could run OrthoFinder and look at the distribution of OG membership size, for diploid assemblies it peaks at 2, but you might find a peak at 2 and 4 for Eubasilissa if it is tetraploid.
- Line 167: how many contaminant contigs were identified, and where did they come from?
- Line 168: the coverage for both species is roughly the same, but the species with the much larger genome is the more contiguous one - any ideas why this is the case?
- Line 184: maybe this is a silly question, but how do you know they are full-length? Based on the B. mori BAC sequence?
- Line 192: a unit for molecular weight, Da?
- Line 224: would be useful to know how many genes are in the Insecta core BUSCO db (i.e., where the 95% comes from). 
- Line 233: is there a possibility that RepetModeler has also classified the repeat-rich fibroin genes as 'repeats', and so these are masked in the assemblies?
- Line 243: this is a huge difference in gene number! Why? Is the E. regina assembly actually a diploid assembly? Or ploidy > 2? [See above comment on kmer plots]. 
- Line 265: "insects have generally been neglected with respect to genome sequencing efforts" - quite a bold statement and I'm not sure I agree, there has been a huge focus on lepidopteran genomics and much of the early sequencing from initiatives such as Darwin Tree of Life have been on insects (also i5k). 
- Line 457: Table 2: any idea why the P. interpunctella HiFi assembly is ~60 Mb shorter than the two Illumina assemblies?
- Line 475: Figures 2 and 3: these are nice figures but I don't quite follow what the two coloured panels on the left are showing, specifically, why are there two panels? A bit more clarification in the legend needed perhaps. 
- Line 476: N and C capitalisedRecommendationMinor Revision

---

## [Reviewer Report]

Reviewer name and names of any other individual's who aided in reviewer Martin PippelDo you understand and agree to our policy of having open and named reviews, and having your review included with the published papers. (If no, please inform the editor that you cannot review this manuscript.)YesIs the language of sufficient quality?YesPlease add additional comments on language quality to clarify if needed
Are all data available and do they match the descriptions in the paper? YesAdditional CommentsAre the data and metadata consistent with relevant minimum information or reporting standards? See GigaDB checklists for examples <a href="http://gigadb.org/site/guide" target="_blank">http://gigadb.org/site/guide</a>YesAdditional CommentsIs the data acquisition clear, complete and methodologically sound?YesAdditional CommentsIs there sufficient detail in the methods and data-processing steps to allow reproduction?YesAdditional CommentsIs there sufficient data validation and statistical analyses of data quality? YesAdditional CommentsIs the validation suitable for this type of data?YesAdditional CommentsIs there sufficient information for others to reuse this dataset or integrate it with other data?YesAdditional Comments(partly) : To make the study fully reproducible the authors need to upload the PacBio HiFi data (e.g. to NCBI). Otherwise the genome assemblies cannot be reproduced with the available raw data in GenBank.Any Additional Overall Comments to the AuthorThe manuscript entitled “Long-read HiFi sequencing correctly assembles repetitive heavy fibroin silk genes in new moth and caddisfly genomes” from Kawahara et al. describes the de novo assembly and gene annotation of two silk-producing insect species Plodia interpunctella and Eubasilissa regina.

The manuscript is well structured and written. Sequencing data, assemblies and genome annotations are publicly available and can be reused by the scientific community. Both contig assemblies show a very high contiguity and good BUSCO scores. Indeed, several from the 118 P. interpunctella and 53 E. regina contigs show telomere repeat sequence at both ends indicating that those represent full chromosomes. Furthermore, the authors showed that even long repetitive genes such as silk fibroin genes were gapless assembled. 

I consider the manuscript as a valuable contribution for the scientific community and do only have some minor comments and suggestions: 

line 129: - which CCS version was used?
line 140: - k-mer length was set to 1? Not 21?
line 148:
- Typo: obd10 reference endopterygota.
- In order to make the Busco scores better comparable to other recent Lepidoptera assemblies it would be better to provide the BUSCO scores for P. interpunctella based on the lepidoptera lineage
line 158: CCS data should be added to GenBank as well. Usually the raw data (subreads.bam) is lossy converted into fastq files from NCBI, which makes it impossible to reproduce the consensus step with pbCCS or even the assembly. 
line 159: Both read coverages are quite high and the heterozygosity rates are with 0.7 (Eubasilissa) and 0.36 (Plodia) high as well. I was wondering if the alternate assemblies were also of a decent quality and if those are published as well? 
line 265: As of today, there are at least 3 other HiFi assemblies available: (GCA_917563855.2, GCA_929108145.1, GCA_917880885.1)
line 457: Table 2 states that E.regina was assembled into 53 contigs. However the assembly available at NCBI GCA_022840565.1 has 123 contigs!? 
RecommendationMinor Revision